# Unsupervised Skill Discovery via Recurrent Skill Training

**Zheyuan Jiang**[1]*    **Jingyue Gao**[2]*    **Jianyu Chen**[1,3]

[1] Institute for Interdisciplinary Information Sciences, Tsinghua University
[2] Department of Computer Science, Tsinghua University
[3] Shanghai Qizhi Institute

## Abstract

Being able to discover diverse useful skills without external reward functions is beneficial in reinforcement learning research. Previous unsupervised skill discovery approaches mainly train different skills in parallel. Although impressive results have been provided, we found that parallel training procedure can sometimes block exploration when the state visited by different skills overlap, which leads to poor state coverage and restricts the diversity of learned skills. In this paper, we take a deeper look into this phenomenon and propose a novel framework to address this issue, which we call Recurrent Skill Training (ReST). Instead of training all the skills in parallel, ReST trains different skills one after another recurrently, along with a state coverage based intrinsic reward. We conduct experiments on a number of challenging 2D navigation environments and robotic locomotion environments. Evaluation results show that our proposed approach outperforms previous parallel training approaches in terms of state coverage and skill diversity. Videos of the discovered skills are available at https://sites.google.com/view/neurips22-rest.

## 1 Introduction

Recent advances in deep reinforcement learning have shown its promising performance in domains ranging from game playing [2, 3], robotics [4, 5] and recommender systems [6]. These applications of reinforcement learning rely on task-specific reward functions for the agents to successfully accomplish the tasks. However, intelligent creatures can automatically explore the environments and learn diverse useful skills in the absence of external supervision. Such ability is beneficial in a variety of situations. For tasks where rewards are non-trivial to design or where the reward signal is sparse, unsupervised skill discovery approaches can provide intrinsic re-

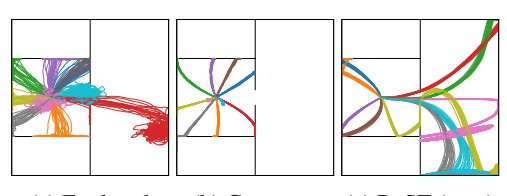

(a) Explored    (b) Converge    (c) ReST (ours)

Figure 1: Skills discovered in a 2D navigation environment.(a) shows the explored states of skills during the training phase of DIAYN [1], where two skills (red and blue) have successfully passed through the bottleneck and explored the states on the right room. (b) shows the states covered by the converged skills of DIAYN, which does not reach states on the right, indicating that the two skills (blue and red) visiting the same state in (a) discourages both skills from visiting that state later. Our proposed method, as shown in (c), successfully reached the states on the right after convergence.

---

*Equal Contribution.

36th Conference on Neural Information Processing Systems (NeurIPS 2022).

wards to help accomplish tasks. Moreover, in hierarchical control problems, unsupervised skill discovery can serve as low-level policies for downstream tasks [7, 1].

Although existing works have shown great potential in discovering diverse useful skills in an unsupervised manner [1, 7, 8], one of the key problems of such methods, as observed in our preliminary experiments and some recent works [9, 10], is that they might suffer from poor state coverage, which may lead to failures in learning desirable useful skills. For instance, in robotic locomotion environments, previous approaches tend to learn 'posing' skills instead of dynamic, far-reaching skills [1]. A straightforward explanation for this phenomenon would be the lack of exploration [9]. However, we argue that this is not the only reason for the poor state coverage. Counterintuitively, we observed that the skills *after convergence* may avoid visiting certain states even if they are explored *during training*. For instance, as shown in our preliminary 2D navigation experiments, the discovered skills fail to cover the states passing through the bottleneck to the right (Figure 1b), even if they have been explored during training (Figure 1a). We call this phenomenon *Exploration degradation*.

In this paper, we take a deeper look into the above phenomenon and show that it is mainly caused by the *parallel* training paradigm, which is a common choice for most existing works [1, 7, 8, 10]. When multiple skills trained in parallel have visited the same state, such state will be prevented from being visited again. Detailed analysis is provided in Section 3.1. Based on the analysis, we propose **Re**current **S**kill **T**raining (ReST), an unsupervised skill discovery algorithm that addresses the exploration degradation issue. Instead of training all the skills in parallel, ReST trains the skills one after another in a recurrent fashion, along with an intrinsic reward that discourages covering frequently visited states of other skills. A preliminary result of using ReST is shown in Figure 1c, where the exploration degradation problem is eliminated and the converged skills are able to visit the space in the right room. Evaluation results on complex 2D navigation and robot locomotion tasks show that our approach can achieve better state coverage and skill divergence compared to baselines.

Our contributions are summarized as follows:

- We discover a new phenomenon that reduces state coverage called exploration degradation, which indicates that some certain states are discouraged from being visited by the learned skills, even if they have been explored during training.

- We show that the main reason causing exploration degradation is that multiple skills visiting the same states can reduce the MI reward in the parallel training paradigm. We then propose **Re**current **S**kill **T**raining (ReST), a recurrent training paradigm along with a state coverage based intrinsic reward, which prevents multiple skills from visiting the same states and alleviates the exploration degradation issue.

- We conduct experiments on various 2D navigation tasks and robot locomotion tasks. Evaluation results show that our method achieves better state coverage and divergence compared to baseline methods. Moreover, ReST learns diverse meaningful robot locomotion skills that have not been shown in previous works.

## 2 Preliminaries

### 2.1 Markov Decision Process

Markov decision process (MDP) can be used to find a reward maximizing policy. It is represented by a tuple $(\mathcal{S}, \mathcal{A}, \mathcal{P}, R, \gamma, \mu)$, where $\mathcal{S}, \mathcal{A}$ are the state and action spaces. $\mathcal{P} : \mathcal{S} \times \mathcal{A} \times \mathcal{S} \to [0, 1]$ is the transition dynamic that maps the state and action into a probability distribution over the next state. $R : \mathcal{S} \times \mathcal{A} \to \mathbb{R}$ is the reward function. $\gamma$ is the discount factor for the reward function while $\mu : \mathcal{S} \to [0, 1]$ is the initial state distribution. The expected discounted cumulative reward can be formulated as $J_R(\pi) = \mathbb{E}_{\tau \sim \pi}[\sum_{t=0}^{\infty} \gamma^t R(s_t, a_t)]$. Thus the overall optimization problem can be written as $\pi^* = \arg\max_{\pi} J_R(\pi)$.

### 2.2 Unsupervised Skill Discovery

Generally speaking, unsupervised skill discovery aims to find a family of skills conditioned on latent $z$, which results in a latent-conditioned policy $\pi(a|s, z)$ that maximizes the mutual information between state and latent:

$$I(S; Z) = H(Z) - H(Z|S) \tag{1}$$
$$= H(S) - H(S|Z) \tag{2}$$

where $s \in \mathcal{S}$, $a \in \mathcal{A}$, $z \in \mathcal{Z}$ are the state, action and latent respectively. Let $(S, Z) \sim p(s, z)$ as the random variables of the state distribution and the latent distribution. As suggested by [11], a variational lower bound for Equation (1) can be derived as:

$$I(S; Z) \geq \mathbb{E}_{(z,s) \sim p(z,s)} \left[ \log q_\phi(z|s) - \log p(z) \right] \tag{3}$$

where $q_\phi(z|s)$ is a learned discriminator approximating $p(z|s)$. Such lower bound also exists for Equation (2):

$$I(S; Z) \geq \mathbb{E}_{(z,s) \sim p(z,s)} \left[ \log q_\phi(s|z) - \log p(s) \right] \tag{4}$$

where $q_\phi(s|z)$ is a learned function approximator of $p(s|z)$.

## 3 Recurrent Skill Training

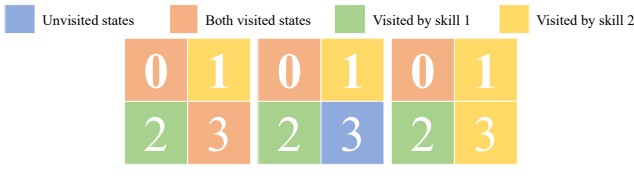

(a) Explored    (b) Parallel    (c) Recurrent

Figure 2: Grid world example. The orange grids are states explored by both the two skills and the green and yellow grids denotes states explored by skill 1 and 2 respectively. Blue grids are unvisited states. (a) shows the state visitation map during exploration while (b) and (c) shows the state visitation of the final converged policies of parallel and recurrent training paradigms respectively.

In this section, we first introduce the exploration degradation phenomenon of previous skill discovery approaches with parallel training paradigm. Then we present our proposed recurrent skill training method that addresses this issue.

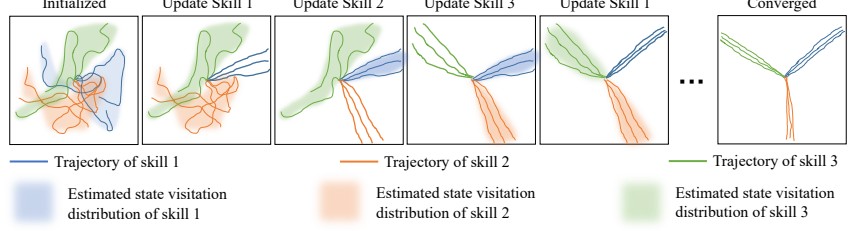

Figure 3: Intuitive illustration of the ReST algorithm. Each color represents a skill. The lines in the figures represent trajectory rollouts of skills whereas the shaded areas indicate the estimated state distribution of the skills. The figure on the left demonstrates the initialization of the three skills, where all skills' trajectories are entangled together. Then ReST starts with training skill 1 by optimizing the reward function in Equation (7), which encourages the trained skill to stay away from the estimated state visitation distributions of other skills. ReST then continues to optimize skill 2, 3 and goes back to optimize skill 1 recurrently. After convergence, the skills would be spread out to cover diverse states as shown in the figure on the right.

---

**Algorithm 1** Recurrent Skill Training

---

**Initialize:** random initialized skill $\pi$, RND networks $\hat{f}$ and $f$
**for** $i = 1$ **to** $N$ **do**
    Set skill $\pi_i = \pi$
    Set RND network $\hat{f}_i = \hat{f}$, $f_i = f$
    Collect on-policy samples with $\pi_i$
    Update RND network $\hat{f}_i$ by minimizing loss in Equation (5)
**end for**
**repeat**
    **for** $i = 1$ **to** $N$ ($N$ is the number of skills) **do**
        **for** $j = 1$ **to** $M$ ($M$ is the number of training epochs for each skill) **do**
            Collect on-policy samples with $\pi_i$;
            Calculate reward using Equation (7);
            Update skill policy $\pi_i$ using any RL algorithms;
            Update RND network $\hat{f}_i$ by minimizing loss in Equation (5) using the latest on-policy samples;
        **end for**
    **end for**
**until** convergence

---

## 3.1 Exploration Degradation

As we observed in our preliminary experiment shown in Figure 1, previous unsupervised skill discovery approaches might suffer from the *exploration degradation* problem, such that some explored states (e.g, states near the bottleneck and in the right room in Figure 1a) are prevented from being visited by the learned skills. We now provide a simplified analysis to show that this phenomenon is mainly caused by the parallel training paradigm commonly used in previous works.

Let us focus on discrete latent case with $N$ different skills. Consider at state $s$, skill $k$ visited state $s$ with $p(z_k|s)$ probability, which is modeled by the discriminator $q_\phi(z_k|s)$. Assume state $s_0$ is only visited by skill $k$ whereas $s_1$ is visited by multiple skills, which means $p(z_k|s_0) = 1$ and $p(z_k|s_1) < 1$. As long as the discriminator $q_\phi$ can model the difference and output $q_\phi(z_k|s_0) > q_\phi(z_k|s_1)$, which is not a strong assumption, the intrinsic reward used by MI approaches would encourage the visitation of state $s_0$ by skill $k$ and discourage visiting $s_1$. This means MI based approaches end up with skills that prefer exploiting states **only** visited by themselves in the previous training epochs, causing exploration degradation.

For clarity, we further explain this issue with a toy example. Consider a $2 \times 2$ grid world as shown in Figure 2. There are four states $0, 1, 2, 3$ and at each state, there are three actions to take: go to the two adjacent states or stay where it is. For simplicity, we use the number of the next state to denote the action, for instance, if the agent is in state $0$ and chooses to go to state $1$, then the action will be denoted as $1$. We use the mutual information described in Equation (1) and assume that the discriminator is perfect: $q_\phi(z|s) = p_\pi(z|s)$. Consider a case when the number of skills is $N = 2$ and during the first collection of samples, skill 1 visited $\{0, 2, 3\}$ while skill 2 visited $\{0, 1, 3\}$, as shown in Figure 2a. Since we have a perfect discriminator, for state 3 the discriminator would output $q_\phi(z_1|3) = q_\phi(z_2|3) = 0.5$, which results in 0 reward for both skills. Therefore, the optimal converged policy for skill 1 would generate trajectory $\{0, 2, 2\}$ while skill 2 would generate trajectory $\{0, 1, 1\}$, as shown in Figure 2b. State 3 is not covered, which is undesirable. This example indicates that even in such an extremely simple case, the exploration degradation phenomenon still exists.

## 3.2 Recurrent Skill Training

Instead of training all skills in parallel, we propose a recurrent training paradigm, along with a state coverage intrinsic reward. We now introduce the details of our proposed method.

**Recurrent Training Paradigm.** As analyzed in Section 3.1, the main reason causing the exploration degradation issue is that the same states are visited by multiple skills in the parallel training paradigm. A natural way to alleviate this issue is to train the skills one after another recurrently. In this way,

we can encourage the latter trained skills to avoid entering the same states covered by the previous skills. Figure 3 illustrates the recurrent training paradigm. Starting with $N$ randomly initialized skills ($N = 3$ in this case), the recurrent training paradigm trains one skill at each epoch whereas the parallel training paradigm trains all the skills together. Furthermore, in order to improve convergence, the recurrent training paradigm updates each skill for $M$ epochs ($M = 2$ in this case) before switching to another skill. In this work, we use $N$ independent neural networks to parametrize the $N$ skills, which can be considered as discrete latent conditioned policies.

**State Coverage Intrinsic Reward.** When implementing the above recurrent training paradigm, the latter trained skill needs to avoid visiting the states frequently visited by other skills. In order to accomplish this objective, we need to identify how frequently each state is visited by each skill, or equally, how novel a given state is to a skill. In this paper, we adopt random network distillation (RND) [12], a simple yet scalable novelty detection approach, to estimate the novelty of a state to a specific skill.

Random network distillation (RND) [12] mainly involves two networks for state novelty detection: a randomly initialized target neural network whose parameters are kept fixed throughout the whole training process and a predictor network trying to fit the target neural network. The target network $f : \mathcal{S} \to \mathbb{R}^k$, where $k$ is the dimension of the output vector, maps a state to a vector while the predictor network $\hat{f} : \mathcal{S} \to \mathbb{R}^k$ is trained to minimize the expected mean square error between the output vector and the target output vector: $||\hat{f}(s) - f(s)||^2$, which distills the target network into the predictor. Therefore, the predictor should have low prediction error on the data it is trained on while have high prediction error on other states. We call such target network and predictor a *pair of RND networks*.

We assign each skill with a pair of RND networks. Each skill's predictor network is trained on the the on-policy data it visited during the last rollout, which means for each skill with latent $z_i \in \mathcal{Z}$, where $i \in \{1, 2, ..., N\}$ is the index of the skill, with state distribution $s \sim p(s|z_i)$, we can train the predictor network using gradient decent by minimizing the mean square error loss $\mathcal{L}_i$

$$\mathcal{L}_i = \mathbb{E}_{s \sim p(s|z_i)} \left[ ||\hat{f}_i(s) - f_i(s)||^2 \right] \tag{5}$$

As mentioned above, the learned predictor network's prediction error $||\hat{f}_i(s) - f_i(s)||^2$ can indicate visitation frequency of a certain state $s$. Intuitively, a higher prediction error indicates higher uncertainty of $\hat{f}_i$ on this state, which further implies its higher novelty. Since we need to avoid visiting states visited by other skills when training a certain skill, a straightforward solution is to define a reward function for skill $i$ with state $s_t$ and action $a_t$ as:

$$r_i(s_t, a_t) = \min_{j \in \{1,2,...,N\}, j \neq i} ||\hat{f}_j(s_{t+1}) - f_j(s_{t+1})||^2 \tag{6}$$

such that states frequently visited by any other skills are less desired. However, this minimum operator would make the reward landscape rugged, which might lead to poor convergence property. To stabilize the training process, we introduce a soft version of the minimum operator in (6):

$$r_i(s_t, a_t) = -\log \left[ \frac{\sum_{j \in \{1,2,...,N\}, j \neq i} e^{\left(-\alpha \cdot ||\hat{f}_j(s_{t+1}) - f_j(s_{t+1})||^2\right)}}{N - 1} \right] \tag{7}$$

where $\alpha$ is a task-specific temperature parameter.

**Practical Algorithm.** We summarize our algorithm in Figure 3 and Algorithm 1. Firstly, we randomly initialize a policy network $\pi_i$ and a pair of RND networks $f_i$ and $\hat{f}_i$ for each skill $i \in \{1, 2, ..., N\}$. Before training skills, ReST first collects on-policy samples for each initialized skill and train their RND networks. Then we train each skill's policy network and the corresponding RND networks for $M$ times during each training epoch. The skill networks are trained recurrently until convergence. For maximizing the intrinsic reward, the ReST algorithm can be combined with an arbitrary reinforcement learning algorithm. In this paper we choose Proximal Policy Optimization (PPO) [13] with generalized advantage estimation (GAE) [14].

**Grid World Example.** We further illustrate the effectiveness of the proposed algorithm using the $2 \times 2$ grid world example with the same setting as in Section 3.1. As suggested by our previous analysis, the parallel training paradigm will end up with two skills that neither of them visits state 3. As mentioned in Section 3.1, the initial two skills visits $\{0, 2, 3\}$ and $\{0, 1, 3\}$ respectively. Here we assume that the RND networks $f_i$ and $\hat{f}_i$ perfectly obtain the state visitation frequency, which means:

$$||\hat{f}_i(s) - f_i(s)||^2 = \left\{ \begin{array}{ll} 0 & \text{if state } s \text{ is visited by skill } i \\ r & \text{otherwise} \end{array} \right. \tag{8}$$

where $r > 0$ is the prediction error. ReST starts by training the corresponding RND networks of each skill and then recurrently train different skills to maximize the intrinsic reward. For skill 1, visiting state 2 would gain reward $r$ while visiting state 1 and 3 would get zero rewards. Therefore, the optimal policy for skill 1 would be visiting $\{0, 2, 2\}$ in sequence. After updating the RND networks of skill 1 correspondingly, for skill 2, visiting state 1 and 3 would gain reward $r$ while visiting 2 would get zero rewards. Therefore, the initial skill 2 is already the optimal policy and the training converges. This way, the optimal trajectory for skill 1 is $\{0, 2, 2\}$ while for skill 2 it is $\{0, 1, 3\}$, as shown in Figure 2c. Therefore, the state space of the $2 \times 2$ grid world is fully co[...] [...] skills.

## 4 Experiments

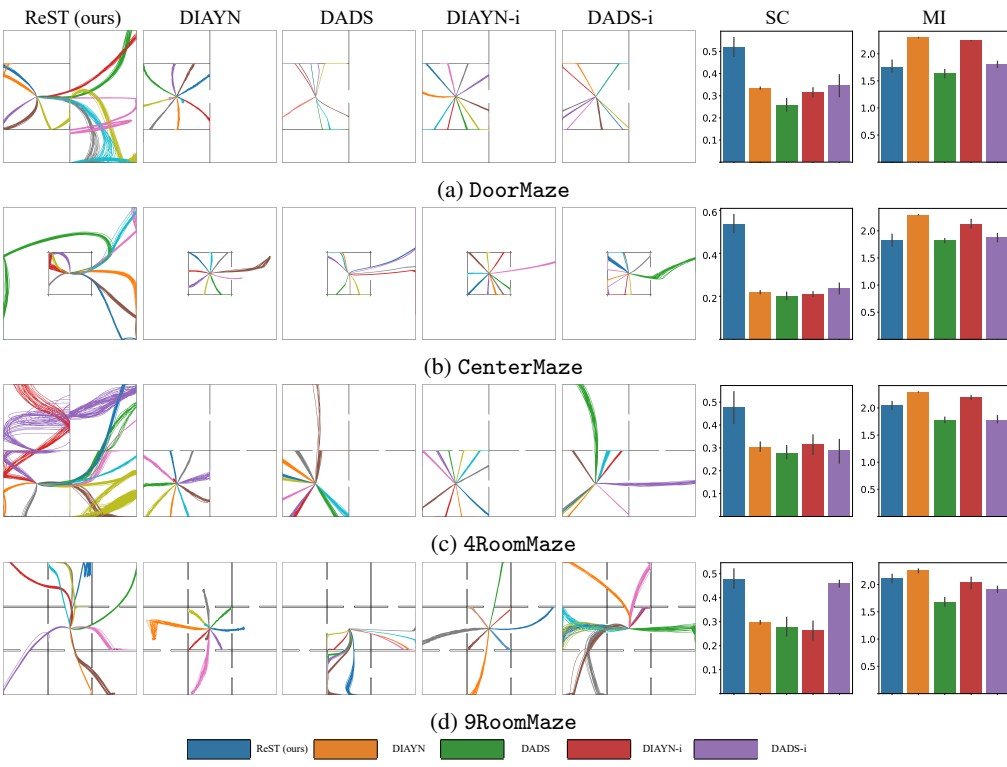

(a) `DoorMaze`

(b) `CenterMaze`

(c) `4RoomMaze`

(d) `9RoomMaze`

Figure 4: Results for 2D navigation experiments. The left five columns of the figure show qualitative results of skills discovered by different algorithms in the corresponding navigation environments. We trained 10 skills for each algorithm where 20 trajectories are rendered for each discovered skill. The right two columns of the figure show the quantitative results of different algorithms. SC stands for state coverage while MI stands for mutual information. SC and MI are calculated based on converged models of each algorithm with three different random seeds. We calculate the SC and MI metrics three times for each converged skill to draw the histogram. Detailed analysis of the evaluation metrics can be found in the Appendix

We conduct experiments on several challenging 2D navigation environments and several robotic locomotion tasks. We compare our proposed approach with two of the most popular unsupervised

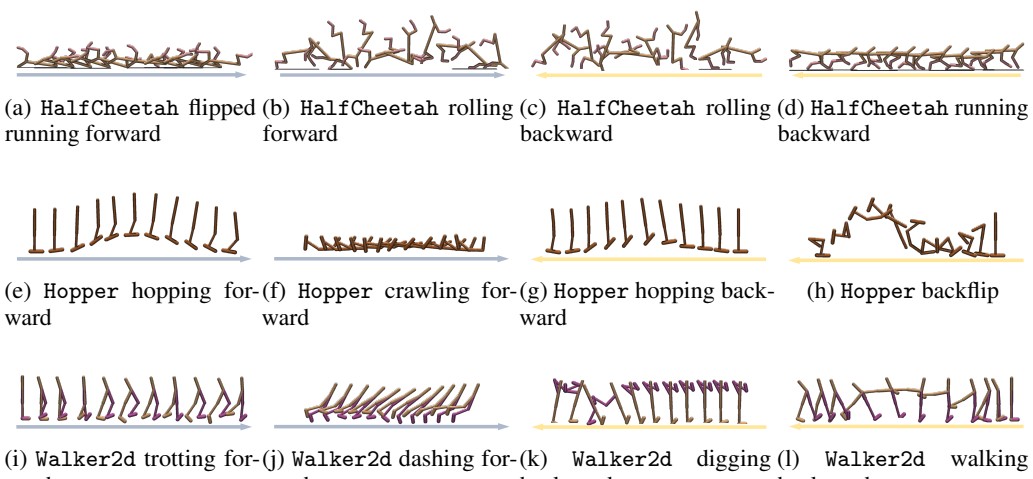

(a) `HalfCheetah` flipped running forward

(b) `HalfCheetah` rolling forward

(c) `HalfCheetah` rolling backward

(d) `HalfCheetah` running backward

(e) `Hopper` hopping forward

(f) `Hopper` crawling forward

(g) `Hopper` hopping backward

(h) `Hopper` backflip

(i) `Walker2d` trotting forward

(j) `Walker2d` dashing forward

(k) `Walker2d` digging backward

(l) `Walker2d` walking backward

Figure 5: Visualization of skills discovered using ReST. The proposed ReST algorithm discovers several high-quality, diverse skills for each robot, including running, flipped running, backflip, dashing, etc. The arrows on the bottom of each sub-figure show the moving direction of each skill.

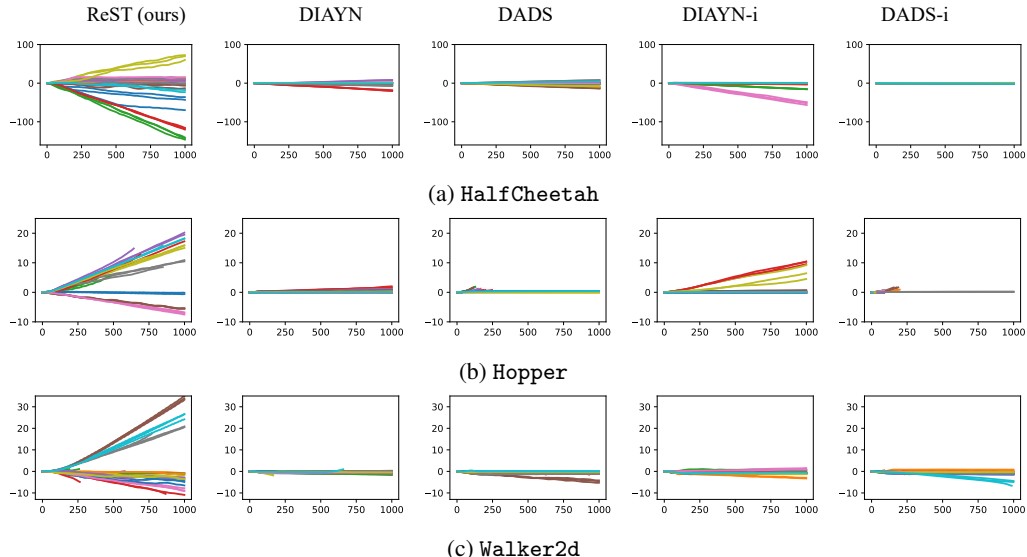

Figure 6: Semi-quantitative results for robotic locomotion tasks. The horizontal axis represents the time-step during evaluation while the vertical axis represents the $x$ position of the robot at each time step. Each color represents a discovered skill and we rollout three trajectories for each skill. Generally speaking, baseline methods discover 'posing' skills instead of dynamic locomotion skills whereas ReST discovers dynamic, far-reaching locomotion skills.

skill discovery approaches, DIAYN [1] and DADS [7]. Both of these approaches use the parallel training paradigm to optimize different skills. Since our proposed approach parameterizes each skill with an independent neural network, we also compare our proposed approach with DIAYN and DADS using independent neural networks, which we call DIAYN-i and DADS-i respectively, for a fair comparison. We present our empirical results both qualitatively and quantitatively. The quantitative results introduce a state coverage metric and a mutual information metric. Our proposed approach significantly outperforms previous skill discovery approaches in terms of state coverage while staying comparable with previous approaches in terms of mutual information. Qualitative results include visualizations of the converged navigation experiments, the state visitation of robotic locomotion tasks, and rendered videos of novel skills discovered using our proposed approach, which can be found in our project website `https://sites.google.com/view/neurips22-rest`. Details of our implementation can be found in the Appendix.

### 4.1 2D Navigation Tasks

**Environments.** We conducted experiments on several challenging 2D navigation environments. The agent is a point mass navigating in a 2D plane with boundaries $[0, 1] \times [0, 1]$. The observation space of the environment has 4 dimensions, including the $x \in [0, 1]$ and $y \in [0, 1]$ position on the plane and the corresponding velocity $v_x$ and $v_y$ belonging to the velocity space $\{(v_x, v_y)|v_x^2 + v_y^2 \leq 0.01\}$. The action space has 2 dimensions, including the acceleration $a_x \in [0, 0.1]$ and $a_y \in [0, 0.1]$ of the point mass agent. There are walls inside the plane and the agent cannot go through the walls. We designed diverse placements of the walls with increasing difficulties for exploring the environments, which could be used to test the effectiveness of our proposed approach.

**Evaluation Metrics.** We introduce two metrics to compare our proposed approach with our comparison baselines. The first one is the state coverage metric. State coverage matters since not covering enough state space might result in failures in learning desirable useful skills. We evaluate the state coverage on the $X$-$Y$ plane by first decomposing the environment into cells and then testing whether each cell is successfully visited. We roll out 1 trajectory for each skill and use the visited states to calculate the state coverage metric. The percentage of cells visited by at least one of the skills is the state coverage. Moreover, merely covering the state space is not enough. The skills need to be informative about which states they are going to visit in the environment so that the skills are meaningful. We use mutual information between skill latent $z$ and the corresponding covered states $s$ to quantify how informative the skill is. Similarly, we decompose the state space into multiple cells and roll out 20 trajectories for each skill to record the probability distribution of states $p(s)$ for all the states visited by the skills. Then we record the probability distribution of states visited by each skill $p(s|z)$ and calculate the entropy $H(S)$ and $H(S|Z_i)$. Since the skill distribution $p(z)$ is a uniform distribution for all skills, we can calculate the mutual information as
$$I(S; Z) = H(S) - \frac{1}{N}\left(\sum_{i \in \{1,2,...,N\}} H(S|Z_i)\right)$$

**Qualitative Results.** We train $N = 10$ skills for each of the algorithms and rollout 20 trajectories to obtain the qualitative results. We conduct experiments on 4 challenging 2D navigation mazes, which we call the `DoorMaze`, `CenterMaze`, `4RoomMaze` and `9RoomMaze`, which are shown in Figure 4. The left 5 columns show the qualitative results. Each rolled-out trajectory is rendered as a curved line and different colors represent different skills. The results indicate that our proposed ReST algorithm can reach more diverse states in the environments whereas baseline approaches like DIAYN or DADS can only reach a small portion of the environments. Usually, the baseline approaches cannot pass through the 'bottlenecks' in the environment. The results support our insight that the parallel training paradigm is one of the causes of previous skill discovery approaches' state coverage issues. Moreover, when using independent neural networks to parameterize different skills, we observe that the performance is not so different from the latent-conditioned parameterization version.

**Quantitative Results.** Beyond qualitative results, we quantify the state coverage and the informativeness of different algorithms using the aforementioned two metrics. As shown in the right two columns of Figure 4, our proposed ReST algorithm significantly outperforms baseline approaches with a parallel training paradigm in terms of state coverage and is comparable with baseline approaches in terms of mutual information $I(S; Z)$. The above results indicate that our proposed approach can cover diverse sets of states without too much sacrifice on informativeness. Quantitative results also evidenced that the difference in performance is not because of the different parameterization since DIAYN and DADS using independent neural networks are not so different from the original ones.

### 4.2 Robotic Locomotion Tasks

We present qualitative results of robotic locomotion skills discovered using our proposed ReST algorithm in MuJoCo [15]. Generally speaking, our proposed ReST algorithm can discover dynamic, far-reaching robotic locomotion skills whereas DIAYN and DADS tend to discover 'posing' skills. Figure 5 shows visualizations of parts of skills discovered using ReST. There are several novel skills discovered, such as `Hopper` backflip, that have not been presented in previous works to the best of our knowledge. More rendered results can be found in the Appendix and our project website `https://sites.google.com/view/neurips22-rest`.

Moreover, we provide semi-quantitative results of the proposed approach. As shown in Figure 6, we draw the agent's $x$ position over timestep, using the skills discovered by ReST and the comparison

baselines. The results are evaluated on `HalfCheetah`, `Hopper` and `Walker2d` tasks. We use the OpenAI Gym [16] settings of the three tasks, where `HalfCheetah` is trained with fixed episode length whereas `Hopper` and `Walker2d` terminate when the agents fall during training. The resulting timestep-$x$ curve indicates that our proposed approach learns skills that are more far-reaching, diverse, and dynamic than baseline methods. This also evidenced that our recurrent training paradigm outperforms the parallel training-based baselines by alleviating their state coverage issues.

## 5   Related Work

**Unsupervised Skill Discovery.** Previous unsupervised skill discovery approaches mainly focus on maximizing the mutual information $I(S; Z)$ to obtain meaningful skills. As discussed in Equation (1) and Equation (2), the mutual information has two forms. Several works follow the Equation (1). VIC [17] uses the mutual information between the final states and the skill latent as the intrinsic reward and optimizes it via reinforcement learning. DIAYN [1] fixes the prior $p(z)$ and uses the mutual information between the skill and its visited states as its intrinsic reward to learn meaningful skills, which has superior performance compared to VIC. VALOR [8] further improves DIAYN by replacing the state-based objective with a trajectory-based objective. DADS [7], on the other hand, uses the Equation (2) form of mutual information and learns a transition model $q_\phi(s'|s, z)$ and uses model predictive control to solve downstream tasks. EDL [9] provides insight into why previous approaches suffer from lack of exploration and proposes an algorithm using a fixed state prior $p(s)$ that to alleviate the issues. IBOL [10] tries to relieve the difficulty of reaching diverse states by introducing a low-level controller. Besides learning a set of skills, SMM [18] formulates skill discovery as a state marginal matching problem and optimizes the KL divergence between the expected state distribution and the current policy's state distribution. MUSIC [19] improves previous unsupervised skill discovery algorithms by adding the mutual information between the surrounding state and the agent state. DDL [20] learns one skill that maximizes the dynamical distance functions of the previous skill. Plan2Explore [21] makes use of self-supervised reward signals to optimize one policy to efficiently gather state-covering dataset. In contrast, ReST optimizes multiple skills and maximize online state coverage. LEXA [22] learns a goal-conditioned policy using self-supervised reward. ReST, on the other hand, does not require the agent to achieve specific goals, which is a more general formulation. Besides the above approaches, there are also other unsupervised skill discovery methods [23, 24, 25, 26].

**Intrinsic Reward.** Another stream of works related to this work is intrinsic reward. In our proposed approach, we use intrinsic reward as an objective to help learn a set of meaningful skills. Intrinsic reward can also help with cases where rewards are sparse by augmenting them to the original reward function. Count-based exploration [27, 28, 29, 30] uses pseudo count to identify the frequently visited states and the less frequently visited ones and adds the count-based bonus as intrinsic rewards to accelerate exploration. Prediction error exploration methods [31, 32, 12, 33, 34] make use of prediction errors as intrinsic rewards based on an insight that states with high prediction error should have higher novelty. Intrinsic reward can also benefit novelty seeking tasks [35, 36], where RSPO [35] made use of an iterative paradigm when training different strategies, which is similar to recurrent training paradigm in this work. Other works augment an information-theoretic intrinsic reward with extrinsic rewards that encourage information gain about the environments [37, 38, 39].

## 6   Discussion

In this paper, we proposed a novel, effective yet simple algorithm called Recurrent Skill Training (ReST). We began by introducing a new phenomenon called exploration degradation which reduces state coverage of the learned skills. We found the key reason for this phenomenon is the parallel training paradigm commonly used in previous skill discovery approaches, such that the same states visited by multiple skills are discouraged from being visited again. Instead of training all skills in parallel at each epoch, ReST trains different skills one after another recurrently. This recurrent training paradigm is supported by an effective prediction error-based intrinsic reward inspired by novelty detection methods. We then conducted experiments on 2D maze navigation to continuous robotic control tasks. Both qualitative and quantitative results show that ReST is able to discover more diverse skills with better state coverage compared to baseline algorithms. Moreover, we demonstrated several novel and dynamic robot locomotion skills that have not been presented in previous works.

There are also some limitations of the proposed algorithm. First of all, compared with previous approaches, our proposed approach has worse sample complexity during skill discovery since only one skill is trained at each epoch, as shown in Figure 3. Moreover, the computational complexity is higher than approaches like [1] or [10] since ReST needs to compute intrinsic reward based on all other skills' prediction errors. Finally, due to the recurrent training paradigm, ReST is currently not scalable to continuous latent, which is in general a better choice as a low-level controller for hierarchical control in downstream tasks. Future works include addressing the above limitations and apply ReST to downstream/hierarchical tasks.

## Acknowledgement

This work is supported by the Ministry of Science and Technology of the People's Republic of China, the 2030 Innovation Megaprojects "Program on New Generation Artificial Intelligence" (Grant No. 2021AAA0150000). We thank Wei Fu and the anonymous reviewers for many insightful suggestions which improved the manuscript.

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
