# OpenReview forum: "Unsupervised Skill Discovery via Recurrent Skill Training"
_NeurIPS.cc/2022/Conference — NeurIPS 2022 Accept_

### Official Review · Reviewer_UvQx · 2022-06-16

**Rating:** 7
**Confidence:** 4
**Soundness:** 4 excellent
**Presentation:** 4 excellent
**Contribution:** 4 excellent

**Summary:**

This paper presents an algorithm, ReST for unsupervised skill discovery. ReST trains different skills in sequential, instead of training all of them in parallel. ReST can address the exploration degradation problem proposed by the authors. The efficiency of the proposed approach is demonstrated in various tasks.

**Questions:**

1. I think the baselines already encourage the diversity of learned skills. Can we develop better approaches to avoid exploration degradation without sacrificing sample efficiency?
2. If the new skill is forced to visit the novel states, it might not able to visit some existing good states, which might be harmful to the exploration. What's your opinion on this?
3. Can the learned skills help hierarchical reinforcement learning?
4. Will the code be released?

**Limitations:**

The authors give a good explanation of the limitations part. However, the broader impact section is not provided.

**Strengths And Weaknesses:**

I tend to give an accept at this moment, and I might change my opinions after reading the rebuttals.
# Strengths:
1. This paper proposes a novel problem in unsupervised skill discovery: the exploration degradation problem. I think this is an interesting problem and is worth an oral presentation for all the researchers in this field to study and understand.
2. The ideas are well introduced and the motivations are quite convincing. I'm convinced by the toy examples provided by the authors. Figure 1 and Figure 2 are very convincing and the ideas are simple and clear.
3. I believe in the effectiveness of sequential skill learning because it can reduce collisions of the learned skills as introduced by the authors.
4. The provided videos of the discovered skills are convincing. They can show that the ReST can learn different skills efficiently.

# Weakness:
1. The proposed videos https://sites.google.com/view/neurips22-rest are not compared to the videos of any baselines. So I cannot evaluate the advantages of the ReST by seeing the video. I wish to see more comparison videos in the rebuttal phase.
2. The proposed recurrent algorithm may consume much longer time than the baselines. So the comparison might be unfair to the baselines. I notice this in the limitation part, so I do not penalize this too much.
3. The quantitative results are not enough. I only see some comparisons on the right of Figure 4. And the comparison cases are quite toy (e.g. in Maze scenarios). The authors should compare to more realistic cases (robotics) and list the numbers for ReST and the baselines. This could benefit future approaches.
4. The writing quality can be improved. Refer to the following section.
5. The code is not provided so it might affect the reproducibility of this work. Although code should not be a must, in the field of reinforcement learning, reproducibility is worth concern without code.
6. The broader impact is not included in this paper. In the standard of Neurips submission now, (expecially this paper is concerned with robotics), this paper needs a broader impact section.

# Writing Suggestions:
1. FIgure 3's text and others are too small for publication quality.
2. Figure 3 does not illustrate the algorithm well. It is too complicated and the authors should walk through Figure 3 (a) in the caption. In general, the ReST algorithm is simple and the authors should provide a simple diagram with clear examples as Figure 3.
3. Eq. 3 to Eq. 7 are a waste of space. You can only leave one form of the information bound and remove other equations.
4. Eq.1 is also a waste of space because it's too trivial.
5. What are N and M in algorithm1? It should be self-contained. You can replace them via "the number of epochs", etc.
6. Some parts are hard to follow. For example, in Ln 131->152, the authors are trying to adopt RND to learn novel skills. However, the descriptions are not organized well so it is hard to understand here if the readers do not know RND before. One suggestion is to discuss the function of the networks here, then dive into details. Another question is, why two networks are used? This question is briefly answered in Ln 147. But I'm not quite convinced by the function.
7. Eq 10 is not illustrated well and there is no ablation study to support this equation.

-------------------------------------
To colleague kKQH,
I agree with you on the strengths parts. For the weakness part, you mainly point out some engineering drawbacks to implementing this paper. I'd like to share my opinions on your concerns:
1. The sample complexity is not that important in this sort of problem. This should be a pure research project and we do not need to deploy the training on a real robot/ industrial application at this moment. The longer training epochs would not cause a longer inference time.
By the way, I think somehow you need that long complexity if you wish the skills are "orthogonal" or "different". Recall the algorithm of Standard Orthogonal in Linear Algebra, we need to find each orthogonal vector at each time to ensure that they are fundamentally different.
2. The continuous part is another long-standing challenge. I don't think they can be solved easily in one framework and many so-called general algorithms can only work well for discrete or continuous problems. This should not be a reason for rejection.
3. I don't quite understand the "catastrophic forgetting of skills", do you mean that the skills can hardly be diverse if you increase the number of skills? This should hold true for all algorithms in this domain, not necessarily a limitation of the proposed approach.
4. The number of networks is fine and one may compress or prune them into one network. This should be an engineering problem.

---

> ### Author Response · Authors · 2022-08-02
> **Response to Reviewer UvQx [1/3]**
>
> ## Response to Reviewer UvQx
>
> We are more than grateful to have your appreciation and we thank you so much for your insightful review and kind suggestions. We are glad to hear that you find the 'exploration degradation' problem interesting, the idea simple and clear, the solution effective and the videos convincing. We hope the following rebuttal successfully address your concerns
>
> ### Response to weaknesses
>
> * **Q**: The proposed videos https://sites.google.com/view/neurips22-rest are not compared to the videos of any baselines. So I cannot evaluate the advantages of the ReST by seeing the video. I wish to see more comparison videos in the rebuttal phase. **A**: Thank you for pointing this out. We will add some rendered videos of our comparison baselines on the project website.
>
> * **Q**: The proposed recurrent algorithm may consume much longer time than the baselines. So the comparison might be unfair to the baselines. I notice this in the limitation part, so I do not penalize this too much. **A**: Thank you for your kind suggestion. In this work, our primary goal is to learn skills that cover the states as much as possible, thus we care more about performance after convergence rather than sample efficiency or training speed. If we view this as a comparison of performance after convergence, it would be a fair comparison. We hope this explaination address your concern.
>
> * **Q**: The quantitative results are not enough. I only see some comparisons on the right of Figure 4. And the comparison cases are quite toy (e.g. in Maze scenarios). The authors should compare to more realistic cases (robotics) and list the numbers for ReST and the baselines. This could benefit future approaches. **A**: Thank you for your suggestion. We will add other quantitative evaluation metrics into the Appendix on more realistic cases like robotics as you mentioned. Thank you for helping us refining our paper.
>
> * **Q**: The writing quality can be improved. Refer to the following section. **A**: We are so grateful that you provided so detailed writing suggestions! Please refer to the following responses.
>
> * **Q**: The code is not provided so it might affect the reproducibility of this work. Although code should not be a must, in the field of reinforcement learning, reproducibility is worth concern without code. **A**: We promisie that we will definitely open source the implementation of our proposed approach immediately upon acceptance.
>
> * **Q**: The broader impact is not included in this paper. In the standard of Neurips submission now, (expecially this paper is concerned with robotics), this paper needs a broader impact section. **A**: Thank you for raising this issue. We revised our paper and included a broader impact section in Appendix G.

---

> ### Author Response · Authors · 2022-08-02
> **Response to Reviewer UvQx [2/3]**
>
> ### Response to questions
>
> * **Q**: I think the baselines already encourage the diversity of learned skills. Can we develop better approaches to avoid exploration degradation without sacrificing sample efficiency? **A**: Thank you for your insightful discussion. In order to answer your question, we need to first find out what is reducing the sample efficiency of ReST. We believe that the reason is mainly caused by the recurrent training paradigm with multiple different networks and buffers. Suppose we have a parallel training paradigm, different skills share the same buffer and each skill can take advantage of other skills' experience and help itself optimize better, which improves sample efficiency whereas in the recurrent training paradigm each skill is trained using only experience of it's own, harming sample efficiency. Furthermore, the parallel training paradigm share the same network weights for different skills, which also improves sample efficiency in cases where multiple skills exist. For instance, if eventually a robot can discover running forward and running backward skills, the two skills share the same 'swing the legs' sub-skill. However in our setting, the two skills need to independently learn from scratch how to swing legs, sacrificing sample efficiency. Therefore, potential solutions include: (1) Keeping a shared replay buffer for different skills to make use of other skills' experience to help optimize the current skill; (2) Using a latent-conditioned neural network for different skills instead of representing each skill using a separate neural network, and adopting techniques form lifelong learning to tackle the catastraphic forgetting issue. Your question might inspire future research in this area and thank you so much for such insightful question!
>
> * **Q**: If the new skill is forced to visit the novel states, it might not able to visit some existing good states, which might be harmful to the exploration. What's your opinion on this? **A**: This is another interesting discussion. Indeed, if the new skill is forced to visit novel states, it might not be able to visit some existing good states. However, the proposed approach does not force new skills to visit novel states, but instead **encourages** skills to visit novel states. For example, when the existing good state leads to following states that are so seldom visited, which the RND networks would generate high rewards that can mitigate the loss of reward by visiting existing states, our algorithm would still encourage visiting the existing good states. This is also evidenced by the empirical result shown in Figure 1( c ) where different skills still visit some same states even though the states are visited by other skills.
>
> * **Q**: Can the learned skills help hierarchical reinforcement learning? **A**: We believe our proposed approach can help with hierarchical reinforcement learning by first applying ReST to discovering diverse skills unsupervised and then training a meta-controller to select the learned skills for downstream tasks. We admit that the performance might be worse than those skill discovery approaches under the continuous latent setting as suggested by [1] and we leave this for future research.
>
> * **Q**: Will the code be released? **A**: The code will definitely be released immediately after acceptance. We will open source our implementation of ReST as soon as this paper get accepted.
>
> [1]Sharma A, Gu S, Levine S, et al. Dynamics-aware unsupervised discovery of skills[J]. arXiv preprint arXiv:1907.01657, 2019.

---

> ### Author Response · Authors · 2022-08-02
> **Response to Reviewer UvQx [3/3]**
>
> ### Response to writing suggestions
>
> * **Q**: FIgure 3's text and others are too small for publication quality. **A**: We have deleted the original Figure 3 and revised according to your next suggestion. Thank you for your suggestion!
>
> * **Q**: Figure 3 does not illustrate the algorithm well. It is too complicated and the authors should walk through Figure 3 (a) in the caption. In general, the ReST algorithm is simple and the authors should provide a simple diagram with clear examples as Figure 3. **A**: We have revised our paper and provided a simple explanation with clear examples. Please inform us if our revision does not meet your requirement. We will be willing to make further modifications.
>
> * **Q**: Eq. 3 to Eq. 7 are a waste of space. You can only leave one form of the information bound and remove other equations. **A**: We have removed Equation 4 and 6 to make the paper more compact. Thank you for your kind suggestion!
>
> * **Q**: Eq.1 is also a waste of space because it's too trivial. **A**: We moved Eq.1 according to your suggestion and thank you for that.
>
> * **Q**: What are N and M in algorithm1? It should be self-contained. You can replace them via "the number of epochs", etc. **A**: $N$ is the number of skills while $M$ is the number of training epoch for each skill. We have modified Algorithm 1 according to your suggestion.
>
> * **Q**: Some parts are hard to follow. For example, in Ln 131->152, the authors are trying to adopt RND to learn novel skills. However, the descriptions are not organized well so it is hard to understand here if the readers do not know RND before. One suggestion is to discuss the function of the networks here, then dive into details. Another question is, why two networks are used? This question is briefly answered in Ln 147. But I'm not quite convinced by the function. **A**: In RND, two networks are trained on the same visited states so that the output of the networks are close on these states, while being different on unvisited states. Therefore the difference of the output of these two networks can be used to estimate how likely a state is visited. We have rewritten this part and maked it easier to follow. Thank you for pointing this out.
>
> * **Q**: Eq 10 is not illustrated well and there is no ablation study to support this equation. **A**: The reason for using Eq 7 (former Eq 10) as the intrinsic reward instead of Eq 6 (former Eq 9) is that the minimum operator would make the training process unstable, leading to poor convergence property. If we take Eq 6 as intrinsic reward, consider the following situation: during the update of skill 1, if skill 2 is the closest to skill 1, then the minimum operator takes skill 2's RND networks' prediction error as the intrinsic reward and optimize the state visitation distribution of skill 1 to be far away from skill 2's. In the next update, skill 1 gets closest to skill 3 and the minimum operator takes skill 3's RND networks' prediction error as the intrinsic reward and optimizes 1 far away from 3. However, this might be problematic since this optimization might take 1 back to where it started, which is close to skill 2. This might cause oscillation and make the training process unstable since it only consider one other skill's state visitation. Eq 7, however, take all other skills' RND networks' prediction error into consideration, releiving the convergence issue of Eq 9.
>
> ### Summary
>
> We notice that you must have spent a lot of time in reading, reviewing and helping us refine our paper and we are so grateful for the suggestions you provide to our paper. The questions you asked, the concerns you raised are not only helping us refine our paper but also inspiring future researches. We hope we have resolved all the concerns you mentioned and we are always willing to address any of your further concerns.
>
> Thank you for your hard work!
>
> Best regards,
>
> The authors

---

> > ### Comment · Reviewer_UvQx · 2022-08-03
> > **Thanks for your response.**
> >
> > Thanks for your response and I think most of my questions are answered well. I hope that you could promise to provide the FULL code of the algorithm by the camera-ready deadline, not providing an empty GitHub repo or providing only the inference code. If so, I believe this paper is above the borderline of Neurips.

---

> > > ### Author Response · Authors · 2022-08-03
> > > **Response to Reviewer UvQx**
> > >
> > > We promise to provide the FULL code of the algorithm immediately after acceptance. Thank you for your appreciation.
> > >
> > > Best regards,
> > >
> > > The authors

---

> > > > ### Comment · Reviewer_UvQx · 2022-08-03
> > > > **Thanks. I don't have further questions at the moment.**
> > > >
> > > > A small tip: you may use the anonymous Github https://anonymous.4open.science/ to send us the code during the rebuttal phase.

---

### Official Review · Reviewer_nWfC · 2022-07-10

**Rating:** 5
**Confidence:** 4
**Soundness:** 3 good
**Presentation:** 4 excellent
**Contribution:** 2 fair

**Summary:**

An ideal scenario of unsupervised skill discovery is to acquire a set of skills that has large state coverage as well as diversity. To this end, this paper suggests optimizing these two criteria by (1) using the state visitation-based reward for exploration and (2) iteratively training skills in that each skill is different from other skills.

The paper identifies that the prior unsupervised skill discovery work discourages skills visiting the bottleneck states due to low intrinsic reward on the bottleneck states $r \sim p(z_i \vert s)$, resulting in the bottleneck state never visited by any skill eventually, which is called the "exploration degradation" problem. This paper claims that this problem comes from simultaneous skill training, and thus proposes to train skills one by one, not in parallel.

The empirical results demonstrate that the proposed method acquires skills that are diverse and identifiable as well as cover wider states, whereas prior works, DIAYN and DADS, only show limited state coverage.

**Questions:**

- Given that RND's prediction networks will become more accurate over time, the reward signal may become less informative about the state visitation statistics. Have the authors experienced this issue for experiments? Is there any way to prevent this?

- Is there any stochasticity in the environment or skills?

### Minor suggestions

- Iteratively sounds more intuitive than recurrently.

- In the abstract, "parallel training procedure inherently discourages exploration" is not intuitive. It would be great to briefly provide an intuition.

- Figure 1 clearly explains the motivation of the paper. However, it does not have enough information to get the intuition without reading until Section 3. A bit more explanation about "two skills (blue and red) visiting the same state (a) discourages both skills to visit that state later (b)" and "colors of trajectories represent skills" would help understanding it.

**Ethics Review Area:**

["I don’t know"]

**Limitations:**

Yes.

**Strengths And Weaknesses:**

### Strengths

- The proposed iterative skill learning scheme is novel under the unsupervised skill discovery setting.

- Experiments on the 2D navigation environments are intuitive and well demonstrate the advantage of the proposed method in terms of diversity and coverage of the learned skills, compared to the baseline methods (in Figure 1 and Figure 4).

- The paper clearly explains its motivation, approach, and experiments. The paper is easy to read.

- Both qualitative and quantitative results are well designed and explained.

### Weaknesses

- There is another direction of work for acquiring a goal-conditioned skill policy in an unsupervised manner [1, 2], which demonstrates its wide state coverage and applicability to complex locomotion as well as manipulation environments. Comparisons to these baselines would be necessary as these works have the same goal of covering a wider range of states but are not limited to a fixed number of discrete skills.

- The proposed method mainly involves two changes from prior work: (1) iterative training for the diversity of skills and (2) the state visitation-based reward for exploration. To verify the effect of each component, it would be helpful to have results about the parallel training version of the proposed reward.

- The proposed method is designed only for discrete skills, and scaling to continuous latent skills seems not trivial given the nature of iterative skill training.

[1] Sekar et al. Planning to Explore via Self-Supervised World Models, ICML 2020

[2] Mendonca et al. Discovering and Achieving Goals via World Models, NeurIPS 2021

---

> ### Author Response · Authors · 2022-08-02
> **Response to Reviewer nWfC [1/2]**
>
> We would like to thank you for your insightful and inspiring review. We are glad to hear that you find our paper novel, experiments intuitive and well demonstrated, paper easy to read and results well presented. We hope the following responses address your concerns about our paper.
>
> ### Response to weaknesses
>
> * **Q**: There is another direction of work for acquiring a goal-conditioned skill policy in an unsupervised manner [1, 2], which demonstrates its wide state coverage and applicability to complex locomotion as well as manipulation environments. Comparisons to these baselines would be necessary as these works have the same goal of covering a wider range of states but are not limited to a fixed number of discrete skills. **A**: Thank you for providing useful suggestions. However, we noticed that the two papers you mentioned are different from ours in respect of overall objective and training settings. For the mentioned papers, they only train **one policy** to generate a **dataset** that covers diverse states. ReST, however, trains **multiple skills** to maximize **online state coverage** of the converged policies. Therefore, our evaluation metric is also designed for online state coverage, where the mentioned two papers are suitable for comparison. Nevertheless, we add discussions about the differences between the proposed approach and the mentioned algorithms in the related work section.
>
> * **Q**: The proposed method mainly involves two changes from prior work: (1) iterative training for the diversity of skills and (2) the state visitation-based reward for exploration. To verify the effect of each component, it would be helpful to have results about the parallel training version of the proposed reward. **A**: We actually have conducted ablation studies to verify the effectiveness of (1) and (2) you mentioned in Appendix F. We conducted two experiments, one using parallel training with ReST reward (F.1) and the other using mutual information reward with the recurrent training paradigm (F.2). The results evidenced our insights that both the recurrent training paradigm and the state coverage intrinsic reward contribute to the performance improvements. Due to the page limit, we were not able to include these results in the main paper. We will include the ablation studies on more environments in the Appendix.
>
> * **Q**: The proposed method is designed only for discrete skills, and scaling to continuous latent skills seems not trivial given the nature of iterative skill training. **A**: We would like to thank you for pointing this limitation out. We agree with you that scaling the recurrent training paradigm to continuous latent could be non-trivial. In fact, reviewer UvQx suggests that this is a long-standing challenge and can hardly be solved in one framework. However, we think it is still possible to successfully extend the proposed approach to a continuous latent. Here we provide a proposal. We can sample a number of latents from the prior continuous latent distribution. We then train a policy network conditioned on the sampled latents using the same procedure and reward of ReST. One potential challenge would be catastrophic forgetting. By adopting techniques from lifelong learning, we might be able to overcome this challenge and discover continuous latent conditioned skills.
>
> [1] Sekar et al. Planning to Explore via Self-Supervised World Models, ICML 2020
> [2] Mendonca et al. Discovering and Achieving Goals via World Models, NeurIPS 2021
>
> ### Response to questions
>
> * **Q**: Given that RND's prediction networks will become more accurate over time, the reward signal may become less informative about the state visitation statistics. Have the authors experienced this issue for experiments? Is there any way to prevent this? **A**: Thank you for your insightful question. However, we do not observe this issue in our experiments since we are only using **on policy samples** to train the RND networks. At each training iteration of the RND networks, the RND networks only see samples visited by the latest policy and this will only cover the state visitation distribution of the latest policy. Moreover, to prevent the RND networks from being less informative about the state visitation statistics casued by generalization, we encourage the RND networks to slightly overfit to the state visitation distribution by adding the number of parameters in the RND networks.
>
> * **Q**: Is there any stochasticity in the environment or skills? **A**: There is stochasticity in the MuJoCo environments where the state initialization is stochastic. There is no stochasticity in state initialization in the 2D navigation environments, which means the starting point is kept fixed. The transition dynamics of both the MuJoCo environments and the 2D navigation environments are deterministic. There is stochasticity in the policies. We followed the PPO setting and sample continuous actions over a learned Gaussian distribution.

---

> > ### Comment · Reviewer_nWfC · 2022-08-04
> > **Thanks for the response**
> >
> > I would like to thank the authors for the response. Many of my concerns are addressed.
> >
> > However, my major concern about the comparisons to Plan2Explore [1] and LEXA [2] is still there. The authors claim
> > > For the mentioned papers, they only train one policy to generate a dataset that covers diverse states. ReST, however, trains multiple skills to maximize online state coverage of the converged policies.
> >
> > A policy trained with Plan2Explore and LEXA [1, 2] is goal-conditioned and each goal can be considered as a (continuous) skill. Moreover, as they are learning a goal-conditioned policy, the converged policy can reach all the diverse states it has explored during training. This is exactly the same as the advantage claimed in this paper. The only difference is that the skills discovered by the proposed method are limited to a discrete set of skills while skills from [1, 2] are in the form of (latent) goals.

---

> > > ### Comment · Reviewer_UvQx · 2022-08-05
> > > **Agree with nWfC.**
> > >
> > > Dear colleague nWfC,
> > > I agree with you that this paper should be compared to Plan2Explore [1] and if I were the authors, I would definitely do the comparisons to make this paper stronger.
> > > Although I think this paper should be novel, the comparisons can better help understand their effectiveness.
> > > Moreover, I think the discussion from the authors is not deep enough, as commented by nWfC. The difference between this paper and [1, 2] should not be the discrete/continuous skill representations. The recurrent parts of this paper and the self-supervised components in [1] should be discussed and compared. In fact, probably they can be combined or they are suited for different situations.
> > > Yours,
> > > Reviewer UvQx

---

> > > ### Author Response · Authors · 2022-08-05
> > > **Thanks for your suggestion**
> > >
> > > Dear Reviewer nWfC:
> > >
> > > Thank you for your suggestion! We think the main differences between ReST and the goal-conditioned methods such as LEXA are: (1) Goal-conditioned methods require the agent to achieve specific goals, which is a more specific problem formulation since not all agent behaviors can be described goal conditioned. For example, the MuJoCo experiments in LEXA are all about posing tasks such as “stand” and “balance” (Fig.2 and Fig.A.1 in LEXA paper), which is because LEXA uses images to represent goals and images alone cannot describe dynamic behaviors such as running, while in our experiments we can learn a lot of continuous locomotion skills besides posing tasks. Such posing behaviors are harmful for state coverage as claimed in our paper, and ReST can largely alleviate this problem; (2) Goal-conditioned methods need a good goal sampling mechanism to indirectly encourage the learned policy covering more of the state space, while ReST requires simply to train each skill recurrently. Neverthless, we agree that it is still valueble to compare them if we view policies conditioned on different goals as different skills to cover diverse states. Therefore we are currently conducting experiments for LEXA with the code open-sourced by the authors and applying it to our environmental settings to compare. However, we believe Plan2Explore does not learn a goal-conditioned policy. In the exploration phase, Plan2Explore learns a single exploration policy jointly with a world model to maximize a novelty-seeking intrinsic reward and generates a dataset containing explored states. Then in the adaptation phase, Plan2Explore learns a single task policy to optimize the given reward function for each target task. Neither of the exploration policy nor the task policy are goal-conditioned, which should not have the ability to cover diverse states as our learned multiple skills, therefore we think it is unfair to compare them. Thanks are for your suggestion and we are always willing to resolve all your further concerns!
> > >
> > > Best regards,
> > >
> > > The authors

---

> > > ### Author Response · Authors · 2022-08-08
> > > **Welcome to check out our new results about LEXA on our project website!**
> > >
> > > Dear reviewer nWfC and reviewer UvQx:
> > >
> > > We would like to thank you for your valuable suggestion about adding LEXA as a comparison baseline. We used the official implementation of LEXA and modified the code to fit into our experiment settings. We changed the CNN encoder and decoder model into MLP to be compatible with our low-dimensional state setting. We kept the same hyper-parameter as the original LEXA paper and used the MLP with the same architecture as ours in the other experiments (2 hidden layers, [64, 64] hidden size). We trained LEXA and evaluated the performance of LEXA using the same setting as other baselines and sampled 10 goals for LEXA in our environment Center Maze. However, the empirical results show that LEXA does not perform well for our task. Please check out the results on our project website (https://sites.google.com/view/neurips22-rest). We are running experiments on other seeds and environments to get more results of LEXA, which will be included in the paper along with the ablation studies soon.
> > >
> > > Here, we would like to share our opinion and analysis of why LEXA does not perform well in our environment. The LEXA framework does not work in our settings mainly because of it’s goal-conditioned setting. The LEXA framework takes in an image as a goal and the intrinsic reward for the achiever $\pi^g$ is encouraging the agent to go to and **stay at** the goal. This is helpful for many real-world robotic tasks, for instance, pick and place manipulation. However, despite the effectiveness, this goal based setting introduces some limitations in other domains. For instance, the goal based intrinsic reward is one of the reasons why the DeepMind Control experiments demonstrated by LEXA contain mainly ‘posing’ skills, since the reward encourages the agent to stay at the state provided by the goal image. This setting is more problematic in our setting since the observation space of the 2D navigation environment is a vector composed of the position in the $x$-$y$ plane and velocities $v_x$, $v_y$. If the achiever is encouraged to stay at the goal, suppose it reached the goal position and velocity at a certain timestep, then in the next time step, if $v_x$ and $v_y$ are not 0, the agent would run away from the goal state immediately, which makes the agent get low reward. This kind of scenario is common in robotics. For instance, if we use the LEXA framework in the legged locomotion tasks, the observation space would contain joint positions and joint velocities, which might bring trouble to the training process. We believe this is one of the reasons why LEXA does not work in our low dimensional state settings. Our proposed approach ReST, however, does not have such issues since it directly learns a set of skills and no explicit goal is given for the agent to reach and stay during the training process.
> > >
> > > We would like to thank you for your hard work and your valuable suggestions during the rebuttal session. Your suggestions are not only making our paper stronger but also inspiring our future research. Moreover, we would appreciate it if you could raise score accordingly if we have addressed your concerns. Thank you again for your hard work!
> > >
> > > Best regards,
> > >
> > > The authors.

---

> > > ### Author Response · Authors · 2022-08-09
> > > **We updated the paper and the supplementary materials**
> > >
> > > Dear reviewer nWfC and reviewer UvQx:
> > >
> > > We have uploaded the revised paper and appendix. Please refer to Appendix F.1, F.2 and F.3 for detailed empirical results and analysis.
> > >
> > > Thank you for helping us making our paper a stronger submission!
> > >
> > > Best regards,
> > >
> > > The authors

---

> ### Author Response · Authors · 2022-08-02
> **Response to Reviewer nWfC [2/2]**
>
> ### Response to minor suggestions
>
> * **Q**: Iteratively sounds more intuitive than recurrently. **A**: Thank you for your kind suggestion. We actually thought about using iterative instead of recurrent in the beginning but iterative could not picture the 'recurrent nature' of the proposed approach since after all skills are updated in one round, we go back to optimize the first skill. Therefore, we prefer to use recurrent instead of iterative.
>
> * **Q**: In the abstract, "parallel training procedure inherently discourages exploration" is not intuitive. It would be great to briefly provide an intuition. **A**: We have changed the expression to "parallel training procedure can sometimes block exploration when the state visited by different skills overlap". We are grateful for your kind suggestion and please inform us if you still have concerns on this expression.
>
> * **Q**: Figure 1 clearly explains the motivation of the paper. However, it does not have enough information to get the intuition without reading until Section 3. A bit more explanation about "two skills (blue and red) visiting the same state (a) discourages both skills to visit that state later (b)" and "colors of trajectories represent skills" would help understanding it. **A**: We are more than grateful to have your suggestion on revising the paper. Your suggestion will make Figure 1 more intuitive to understand and we have revised our paper according to your suggestion.
>
> ### Summary
>
> We would like to thank you again for your responsible review and valuable suggestions. We hope we have resolved all the concerns you mentioned and we deeply appreciate that if you could consider raising the score accordingly. We are always willing to address any of your further concerns.
>
> Thank you for your hard work!
>
> Best regards,
>
> The authors

---

### Official Review · Reviewer_kKQH · 2022-07-11

**Rating:** 4
**Confidence:** 5
**Soundness:** 4 excellent
**Presentation:** 4 excellent
**Contribution:** 2 fair

**Summary:**

Parallel training of skill discovery methods results in failure of learning some skills. This happens because certain states are never visited by the learned skills. Authors propose recurrent skill training, which involves training a particular skill for some epochs followed by the next skill for the next few epochs.
They further shape the reward according to the state coverage. This prevents multiple skills from visiting the same states. Further experiments on various 2D navigation tasks and robot locomotion tasks show the new method achieves better state coverage and divergence compared to baseline methods.


**Questions:**

How does one decide how many epochs to train a particular skill?

Given the current weakness it is difficult to accept the paper. If the authors provide good reasons why this work is important despite the weakness, I am willing to change my score.


**Limitations:**

Please address catastrophic forgetting when number of skills gets large.

**Strengths And Weaknesses:**

Strengths:
The paper finds a clear and important weakness in the current skill discovery methods. And solves it with a "somewhat" novel solution.

The paper is well written and clearly explained. The experiments are also well done.

Weakness:
The paper does come up with a solution to a problem. But the proposed solution creates more problems of its own.
1) Proposed method has worse sample complexity compared to other baselines (while other methods have less state coverage)
2) Does not work with a continuous latent
3) Does not address catastrophic forgetting of skills as number of skills grow
4) Needs at least 2*N networks, where N is the number of skills


Post-Rebuttal:
When we compare ReST against current methods in unsupervised skill learning, ReST falls short in multiple accounts.

Has higher sample and computational complexity compared to other methods
Cannot handle continuous latent (while other comparative methods can)
Too many moving pieces compared to other methods (Multiple networks for skills, for novelty detection)

Thus, it is difficult for me to recommend acceptance of the paper. I thank you for your efforts and response.

---

> ### Author Response · Authors · 2022-08-02
> **Response to Reviewer kKQH**
>
> Thank you for your constructive feedback. We are delighted that you find our paper well written, easy to read and our experiments well done. We hope the following rebuttal address your concerns.
>
> ### Response to weaknesses
>
> * **Q**: Proposed method has worse sample complexity compared to other baselines (while other methods have less state coverage). **A**: Thank you for pointing out this limitation of our proposed approach. We admit that the proposed approach has worse sample complexity compared to some state-of-the-art baselines. However, there are many domains where sample efficiency are becoming less and less important. For instance, in robotics, massively parallel GPU accelerated simulation is available (e.g. Isaac Gym[1], Brax[2]), making samples cheap to obtain and reducing training time from days to minutes [3]. Moreover, we aim to train highly diverse skills that are different from each other as much as possible, which inherently require much more training samples. This is analogous to the example given by reviewer UvQx, which suggests that the algorithm of Standard Orthogonal in Linear Algebra requires finding each orthogonal vector at each time to ensure that they are fundamentally different. We sincerely hope you take the points we made into consideration.
>
> * **Q**: Does not work with a continuous latent. **A**: Thank you for pointing our this limitation. Although in this work ReST only works with discrete "latent", it is possible to extend the proposed approach to a continuous latent. Here we provide a proposal. We can sample a number of latents from the prior continuous latent distribution. We then train a policy network conditioned on the sampled latents using the same procedure and reward of ReST. One potential challenge would be catastrophic forgetting. By adopting techniques from lifelong learning, we might be able to overcome this challenge and discover continuous latent conditioned skills. However, this could be non-trivial. In fact, as suggested by reviewer UvQx, the continuous part is another long-standing challenge and cannot be solved easily in one framework. We leave this challenge for future research.
>
> * **Q**: Does not address catastrophic forgetting of skills as number of skills grow. **A**: Thank you for raising this concern, but we believe there are some misunderstanding. Catastrophic forgetting is the main challenge for lifelong learning, but we here avoid this issue by using multiple neural networks to represent different skills and the catastrophic forgetting problem does not exist in our current proposed approach. However, this concern you raised is inspiring. For example, if we want to represent all skills with one latent-conditioned neural network, we might suffer from catastrophic forgetting. In this case, we might need to adopt techniques from lifelong learning to address such issue.
>
> * **Q**: Needs at least $2\times N$ networks, where $N$ is the number of skills. **A**: Thank you for pointing this out. Here we propose two potential ways to address this issue. First, we may extend ReST to work with one latent conditioned network by adopting techniques from lifelong learning and train multiple skills sequentially, as mentioned in the previous response. Second, as suggested by reviewer UvQx, we can distill different skills into one neural network. We hope these two potential approaches address your concern and thank you for inspiring future research.
>
> [1]Makoviychuk V, Wawrzyniak L, Guo Y, et al. Isaac gym: High performance gpu-based physics simulation for robot learning[J]. arXiv preprint arXiv:2108.10470, 2021.
> [2]Freeman C D, Frey E, Raichuk A, et al. Brax--A Differentiable Physics Engine for Large Scale Rigid Body Simulation[J]. arXiv preprint arXiv:2106.13281, 2021.
> [3]Rudin N, Hoeller D, Reist P, et al. Learning to walk in minutes using massively parallel deep reinforcement learning[C]//Conference on Robot Learning. PMLR, 2022: 91-100.
>
> ### Resonse to questions
>
> * **Q**: How does one decide how many epochs to train a particular skill? **A**: This is decided according to empirical evidence. In experiments in our paper, we tried several different training epochs and chose the well performing one to present in our paper. The proposed algorithm is not sensitive to training epochs for each skills and taking $10$ or $20$ should work for most tasks.
>
> ### Summary
>
> We would like to thank you again for your responsible review and valuable suggestions. We hope we have resolved all the concerns you mentioned and we deeply appreciate that if you could consider raising the score accordingly. We are always willing to address any of your further concerns.
>
> Thank you for your hard work!
>
> Best regards,
>
> The authors

---

> > ### Comment · Reviewer_UvQx · 2022-08-03
> > **Comments on the related concerns.**
> >
> > To colleague kKQH, I agree with you on the strengths parts. For the weakness part, you mainly point out some engineering drawbacks to implementing this paper. I'd like to share my opinions on your concerns:
> >
> > The sample complexity is not that important in this sort of problem. This should be a pure research project and we do not need to deploy the training on a real robot/ industrial application at this moment. The longer training epochs would not cause a longer inference time. By the way, I think somehow you need that long complexity if you wish the skills are "orthogonal" or "different". Recall the algorithm of Standard Orthogonal in Linear Algebra, we need to find each orthogonal vector at each time to ensure that they are fundamentally different.
> > The continuous part is another long-standing challenge. I don't think they can be solved easily in one framework and many so-called general algorithms can only work well for discrete or continuous problems. This should not be a reason for rejection.
> > I don't quite understand the "catastrophic forgetting of skills", do you mean that the skills can hardly be diverse if you increase the number of skills? This should hold true for all algorithms in this domain, not necessarily a limitation of the proposed approach.
> > The number of networks is fine and one may compress or prune them into one network. This should be an engineering problem.

---

> ### Author Response · Authors · 2022-08-07
> **Welcome to check out our response and discuss with us!**
>
> Dear reviewer kKQH:
>
> We would like to thank you again for your hard work in reviewing our paper! You pointed out some valuable suggestions about our paper and we made a response trying our best to answer all your questions. We hope we have resolved all the concerns you mentioned and we deeply appreciate that if you could consider raising the score accordingly. If you have any further concerns, we will be more than happy to make a response.
>
> Looking forward to hearing from you soon!
>
> Best regards,
>
> The authors

---

> ### Comment · Reviewer_UvQx · 2022-08-07
> **Have you changed your mind?**
>
> Dear reviewer kKQH:
> Have you changed your mind so far?

---

> ### Author Response · Authors · 2022-08-08
> **We are always willing to address your further concerns!**
>
> Dear reviewer kKQH:
>
> We would like to appreciate again your hard work in reviewing our paper! We are eager to hear about your opinions on the response we made. Have our responses resolved your concerns about our paper? We are always willing to address your further concerns!
>
> Looking forward to hearing from you soon!
>
> Best regards,
>
> The authors

---

### Meta-Review · Area_Chair_nzyU · 2022-08-27

**Recommendation:** Accept
**Confidence:** Less certain

**Metareview:**

The paper identifies a problem in prior works on skill-learning: some states that are visited in training by diverse skills may not be visited after skills are learned (dubbed as exploration degradation problem). The problem is well-motivated and is an important one in unsupervised skill learning. Authors then propose a method to overcome the identified issue. Comparisons are made to state-of-the-art methods such as DADS and LEXA.

The reviewers are split in their opinion: UvQx recommends an accept, nWfC recommends borderline accept, while kkQh recommends rejecting the paper. Authors addressed the primary concern of nWfC on comparison with LEXA. kkQh's main concerns are about the solution not being elegant. While I agree that more elegant solutions can be found, I think that identifying a bottleneck in learning diverse skills is of good value to the community. Furthermore, the proposed solution works across a range of environments. Therefore, I am lean towards a positive recommendation for this paper. I encourage the authors to clearly address the reviewers suggestions and comments in the camera ready version.

**Award:**

No

---

### Decision · Program_Chairs · 2022-09-14

Accept